# Inter/Intra-Observer Agreement in Video-Capsule Endoscopy: Are We Getting It All Wrong? A Systematic Review and Meta-Analysis

**DOI:** 10.3390/diagnostics12102400

**Published:** 2022-10-02

**Authors:** Pablo Cortegoso Valdivia, Ulrik Deding, Thomas Bjørsum-Meyer, Gunnar Baatrup, Ignacio Fernández-Urién, Xavier Dray, Pedro Boal-Carvalho, Pierre Ellul, Ervin Toth, Emanuele Rondonotti, Lasse Kaalby, Marco Pennazio, Anastasios Koulaouzidis

**Affiliations:** 1Gastroenterology and Endoscopy Unit, University Hospital of Parma, University of Parma, 43126 Parma, Italy; 2Department of Clinical Research, University of Southern Denmark, 5230 Odense, Denmark; 3Department of Surgery, Odense University Hospital, 5000 Odense, Denmark; 4Department of Gastroenterology, University Hospital of Navarra, 31008 Pamplona, Spain; 5Center for Digestive Endoscopy, Sorbonne University, Saint Antoine Hospital, APHP, 75012 Paris, France; 6Gastroenterology Department, Hospital da Senhora da Oliveira, Creixomil, 4835 Guimarães, Portugal; 7Division of Gastroenterology, Mater Dei Hospital, 2090 Msida, Malta; 8Department of Gastroenterology, Skåne University Hospital, Lund University, 20502 Malmö, Sweden; 9Gastroenterology Unit, Valduce Hospital, 22100 Como, Italy; 10University Division of Gastroenterology, City of Health and Science University Hospital, University of Turin, 10126 Turin, Italy; 11Department of Medicine, OUH Svendborg Sygehus, 5700 Svendborg, Denmark; 12Surgical Research Unit, OUH, 5000 Odense, Denmark; 13Department of Social Medicine and Public Health, Pomeranian Medical University, 70-204 Szczecin, Poland

**Keywords:** capsule endoscopy, video reading, agreement, small bowel, colon

## Abstract

Video-capsule endoscopy (VCE) reading is a time- and energy-consuming task. Agreement on findings between readers (either different or the same) is a crucial point for increasing performance and providing valid reports. The aim of this systematic review with meta-analysis is to provide an evaluation of inter/intra-observer agreement in VCE reading. A systematic literature search in PubMed, Embase and Web of Science was performed throughout September 2022. The degree of observer agreement, expressed with different test statistics, was extracted. As different statistics are not directly comparable, our analyses were stratified by type of test statistics, dividing them in groups of “None/Poor/Minimal”, “Moderate/Weak/Fair”, “Good/Excellent/Strong” and “Perfect/Almost perfect” to report the proportions of each. In total, 60 studies were included in the analysis, with a total of 579 comparisons. The quality of included studies, assessed with the MINORS score, was sufficient in 52/60 studies. The most common test statistics were the Kappa statistics for categorical outcomes (424 comparisons) and the intra-class correlation coefficient (ICC) for continuous outcomes (73 comparisons). In the overall comparison of inter-observer agreement, only 23% were evaluated as “good” or “perfect”; for intra-observer agreement, this was the case in 36%. Sources of heterogeneity (high, I^2^ 81.8–98.1%) were investigated with meta-regressions, showing a possible role of country, capsule type and year of publication in Kappa inter-observer agreement. VCE reading suffers from substantial heterogeneity and sub-optimal agreement in both inter- and intra-observer evaluation. Artificial-intelligence-based tools and the adoption of a unified terminology may progressively enhance levels of agreement in VCE reading.

## 1. Introduction

Video-capsule endoscopy (VCE) entered clinical use in 2001 [1]. Since then, several post-market technological advancements followed, making capsule endoscopes the prime diagnostic choice for several clinical indications, i.e., obscure gastrointestinal bleeding (OGIB), iron-deficiency anemia (IDA), Crohn’s disease (diagnosis and monitoring) and tumor diagnosis. Recently, the European Society of Gastrointestinal Endoscopy (ESGE) endorsed colon capsule endoscopy (CCE) as an alternative diagnostic tool in patients with incomplete conventional colonoscopy or contraindication for it, when sufficient expertise for performing CCE is available [2]. Furthermore, the COVID-19 pandemic has bolstered CCE (and double-headed capsules) in clinical practice as the test can be completed in the patient’s home with minimal contact with healthcare professionals and other patients [3,4].

The diagnostic yield of VCE depends on several factors, such as the reader’s performance, experience [5] and accumulating fatigue (especially with long studies) [6]. Although credentialing guidelines for VCE exist, there are no formal recommendations and only limited data to guide capsule endoscopists on how to read the many images collected in each VCE [7,8]. Furthermore, there is no guidance on how to increase performance and obtain a consistent level of high-quality reporting [9]. With accumulating data on inter/intra-observer variability in VCE reading (i.e., degree of concordance between multiple readers/multiple reading sessions of the same reader), we embarked on a comprehensive systematic review of the contemporary literature and aimed to estimate the inter- and intra-observer agreement of VCE through a meta-analysis.

## 2. Materials and Methods

### 2.1. Data sources and Search Strategy

We conducted a systematic literature search in PubMed, Embase and Web of Science in order to identify all relevant studies in which inter- and/or intra-observer agreement in VCE reading was evaluated. The primary outcome was the evaluation of inter- and intra-observer agreement in VCE examinations. The last literature search was performed on 26 September 2022. The complete search strings are available in Appendix A. This review was registered at the PROSPERO international register of systematic reviews (ID 307267).

### 2.2. Inclusion and Exclusion Criteria

The inclusion criteria were: (i) full text articles; (ii) articles reporting either inter- or intra-observer agreement values (or both) of VCE reading; (iii) articles in English/Italian/Danish/Spanish/French language. Exclusion criteria were: article types such as reviews, case reports, conference papers or abstracts.

### 2.3. Screening of References

After exclusion of duplicates, references were independently screened by six authors (P.C.V., U.D., T.B.-M., X.D., P.B.-C., P.E.). Each author screened one fourth of the references (title and abstract), according to the inclusion and exclusion criteria. In case of discrepancy, the reference was included for full text evaluation. This approach was then repeated on included references with an evaluation of the full text by three authors (P.C.V., U.D., T.B.-M.). In case of discrepancy in the full-text evaluation, the third author would also evaluate the reference and a consensus discussion between all three would determine the outcome.

### 2.4. Data Extraction

Data were extracted in accordance with the Preferred Reporting Items for Systematic Reviews and Meta-Analyses (PRISMA) [10]. We extracted data on patients’ demographics, indication for the procedure, the setting for the intervention, the type of VCE and its completion rate, and the type of test statistics.

### 2.5. Study Assessment and Risk of Bias

Included studies underwent an assessment of methodological quality by three independent reviewers (P.C.V., U.D., T.B.-M.) through the Methodological Index for Non-Randomized Studies (MINORS) assessment tool [11].

Items 7, 9, 10, 11 and 12 were omitted, as they were not applicable to the included studies; therefore, since the global ideal score for non-comparative studies, in MINORS, is at least two thirds of the total score (n = 24), we applied the same proportion to the maximum score with omitted items (n = 14) obtaining the arbitrary cut-off value of 10.

### 2.6. Statistics

In the included studies, different test statistics were used when reporting the degree of observer agreement. The most common ones are the Kappa statistics for categorical outcomes and the intra-class correlation coefficient (ICC) for continuous outcomes. Kappa and ICC are not directly comparable and our analyses were therefore stratified by type of test statistics.

The Kappa statistics estimates the degree of agreement between two or more readers, while taking into account the chance agreement that would occur if the readers guessed at random. Cohen’s Kappa was introduced in order to improve the previously common used percent agreement [12].

The ICC is a measure of the degree of correlation and agreement between measurements and is a modification of the Pearson correlation coefficient, which measures the magnitude of correlation between variables (or readers) but, in addition, ICC takes readers’ bias into account [13,14].

Less commonly reported were the Spearman rank correlation [15], Kendall’s coefficient and the Kolmogorov–Smirnov test. First, we evaluated each comparison using guidelines for the specific test statistics (Table 1) and divided them into groups of “None/Poor/Minimal”, “Moderate/Weak/Fair”, “Good/Excellent/Strong” and “Perfect/Almost perfect” to report the proportions of each, stratified by inter/intra-observer agreement evaluations.

As no guidelines were identified for the Kendall’s coefficient and the Kolmogorov–Smirnov test, we adopted the guidelines used for Kappa as the scales were similar. The mean value was estimated stratified by test statistic. The significance level was set at 5%, and 95% confidence intervals (CIs) were calculated. All pooled estimates were calculated in random effects models stratified into four categories; inter-observer Kappa, intra-observer Kappa, inter-observer ICC and intra-observer ICC. To investigate publication bias and small study effects, Egger’s tests were performed and illustrated by funnel plots. Individual study data were extracted and compiled in spreadsheets for pooled analyses. Data management was conducted in SAS (SAS Institute Inc. SAS 9.4. Cary, NC, USA), while analyses and plots were performed in R (R Development Core Team, Boston, MA, USA) using the metafor and tidyverse packages [16,17].

## 3. Results

Overall, 483 references were identified from the databases. After the removal of duplicates, 269 were screened, leading to 95 references for full-text reading. One additional reference was retrieved via snowballing. Sixty (n = 60) studies were eventually included, 37 of which had reported information on variance for their agreement measures, enabling them to be included for pooled estimates (Figure 1). MINORS scores ranged from 7 to 14, with the majority of references scoring 10 or above (n = 52) (Table 2).

Regarding the type of statistics used in the 60 included studies, 46 reported Kappa statistics (424 comparisons), 11 reported ICC (73 comparisons), 5 reported Spearman rank correlations (60 comparisons), 2 reported Kendall’s coefficients (20 comparisons) and 1 reported Kolmogorov–Smirnov tests (2 comparisons).

The analysis of combined inter/intra-observer values (overall means) per type of statistics revealed a weak agreement for the comparisons measured by Kappa statistics (0.53, CI 95% 0.51; 0.55), good for ICC (0.81, CI 95% 0.78; 0.84) and moderate for Spearman rank correlation (0.73, CI 95% 0.68; 0.78). For Kendall’s coefficient and Kolmogorov–Smirnov tests, too few studies were identified to make an overall evaluation (Table 3).

The distribution of evaluations, stratified by inter/intra-observer agreements, was analyzed by combining all specific comparisons regardless of the type of statistics models (Kappa alone was considered in 25 inter-observer comparisons, whenever more than one model was applied for the same outcome): in 479 inter-observer comparisons, a “good” or “perfect” agreement was obtained in only 23% of the cases; in 75 intra-observer comparisons, this was the case in 36% of the cases (Figure 2).

For the pooled random effects models stratified by inter/intra-observer and test statistic, the overall estimates of agreement ranged from 0.46 to 0.84, although a substantial degree of heterogeneity was present in all four models (Figure 3 and Figure 4). The I^2^ statistic ranged from 81.8% to 98.1% (Figure 4). Meta-regressions investigating the possible sources of heterogeneity found no significance of any variable for ICC inter-observer agreement, but for Kappa inter-observer agreement, country, capsule type and year of publication may have contributed to the heterogeneity.

For the random effects models of the overall inter/intra-observer agreements, the Eggers tests resulted in *p*-values < 0.01 for inter/intra-observer ICC models, 0.78 for Kappa inter-observer and 0.20 for Kappa intra-observer (Figure 5).

## 4. Discussion

Reading VCE videos is a laborious and time-consuming task. Previous work has showed that the inter-observer agreement and the detection rate of significant findings are low, regardless of the reader’s experience [5,78]. Moreover, attempts to improve performance by a constructed upskilling training program did not significantly impact readers with different experience levels [78]. Fatigue has been blamed as a significant determinant of missed lesions: a recent study demonstrates that reader accuracy declines after reading just one VCE video, and that neither subjective nor objective measures of fatigue were sufficient to predict the onset of the effects of fatigue [6]. Recently, strides were made in establishing a guide for evaluating the relevance of small-bowel VCE findings [79]. Above all, artificial intelligence (AI)-supported VCE can identify abnormalities in VCE images with higher sensitivity and significantly shorter reading times than conventional analysis by gastroenterologists [80,81]. AI has, of course, no issues with inter-observer agreement and is poised to become an integral part of VCE reading in the years to come. AI develops on the background of human-based ‘ground truth’ (usually subjective expert opinion) [82]. So, how do we as human readers get it so wrong?

The results of our study show that the overall pooled estimate for “perfect” or “good” inter- and intra-observer agreement was only 23% and 37%, respectively (Figure 2). Although significant heterogeneity was noted in both Kappa statistic and ICC-based studies, the overall combined inter/intra-observer agreement for Kappa-evaluated outcomes was weak (0.46 and 0.54, respectively), while for ICC-evaluated outcomes the agreement was good (0.83 and 0.84, respectively).

A possible explanation to this apparent discrepancy is that ICC outcomes are more easily quantifiable, therefore providing a higher degree of unified understanding on how to evaluate, whereas categorical outcomes in Kappa statistics may be prone to a more subjective evaluation; for instance, substantial heterogeneity may be caused by pooling observations without unified definition of the outcome variables (e.g., cleansing scale, per segment or patient, categorical subgroups differences). 

A viable solution to the poor inter-/intra-observer agreement on VCE reading could be represented by AI-based tools. AI offers the opportunity of a standardized observer-independent evaluation of pictures and videos relieving reviewers’ workload, but are we ready to rely on non-human assessment of diagnostic examinations to decide for subsequent investigations or treatments? Several algorithms reported with high accuracy have been proposed for VCE analysis. The main deep learning algorithm for image analysis has become convolutional neural networks (CNN) as they have shown excellent performances for detecting esophageal, gastric and colonic lesions [83,84,85]. However, some important shortcomings need to be overcome before CNNs are ready for implementation in clinical practice. The generalization and performance of CNNs in real-life settings are determined by the quality of data used to train the algorithm. Hence, large amounts of high-quality training data are needed with external algorithm validation, which necessitates collaboration between international centers. A high sensitivity from AI should be prioritized even at the cost of the specificity as AI findings should always be reviewed by human professionals. 

This study shows several limitations. As VCE is used for numerous indications and for all parts of the GI tract, an inherent weakness is the natural heterogeneity of the included studies, which is evident in the pooled analyses (I^2^ statistics > 80% in all strata). The meta-regressions indicated that country, capsule type and year of publication may have contributed to the heterogeneity for Kappa inter-observer agreement, whereas no sources were identified in ICC analyses; furthermore, the Eggers’ tests indicated publication bias in ICC analyses but not in Kappa analyses. Therefore, there is a risk that specific pooled estimates may be inaccurate, but the heterogeneity may also be the result of very different ways of interpreting videos or definitions of outcomes between sites and trials. No matter these substantial weaknesses to the results of the pooled analyses, the proportions of agreements and the great variance in agreements are clear. In more than 70% of the published comparisons, the agreement between readers is moderate or worse, as for intra-observer agreement.

Data regarding the reader’s experience were originally extracted but omitted in the final analysis because of heterogeneity of the terminology and of the lack of a unified experience scale. This should not be considered as a problem, as most studies fail to confirm a significant lesion detection rate difference between experienced and expert readers, physician readers and nurses [86,87], while some of them point to possible equalization of any difference between novices and experienced even only after one VCE reading due to fatigue [6].

Moreover, we decided not to perform any subgroup analysis based on possible a priori clustering of findings (e.g., bleeding lesions, ulcers, polyps, etc.); the reason for this choice is related, once again, to the extreme variability of encountered definitions and the lack of a uniform terminology. 

## 5. Conclusions

As of today, the results of our study show that VCE reading suffers from a sub-optimal inter/intra-observer agreement.

For future meta-analyses, more studies are needed enabling strata of subgroups specific to the outcome and indication, which may limit the heterogeneity. The heterogeneity may also be reduced by stratifying analyses based on the experience level of the readers or the number of them in comparisons, as this will most likely affect the agreement. The progressive implementation of AI-based tools will possibly enhance the agreement in VCE reading between observers, not only reducing the ”human bias” but also relieving the significant burden in workload.

## Figures and Tables

**Figure 1 diagnostics-12-02400-f001:**
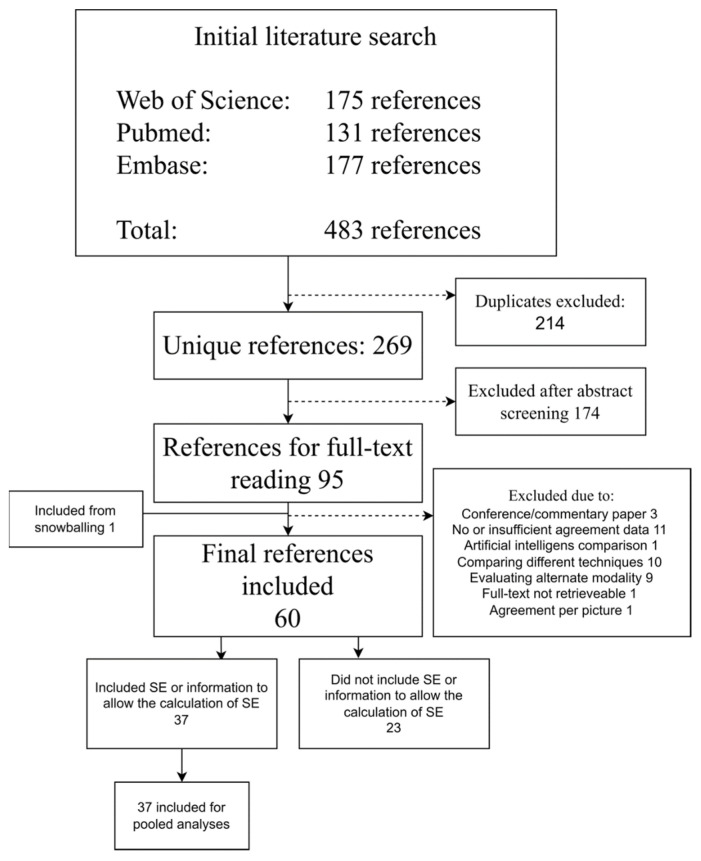
Flow diagram of the study. Abbreviations: SE, standard error.

**Figure 2 diagnostics-12-02400-f002:**
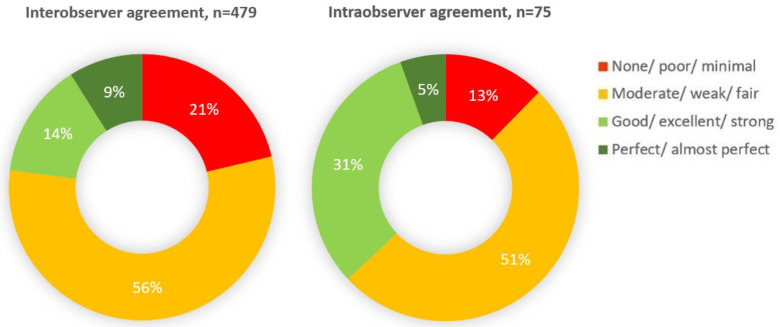
Distribution of agreement evaluations stratified by inter/intra-observer agreements.

**Figure 3 diagnostics-12-02400-f003:**
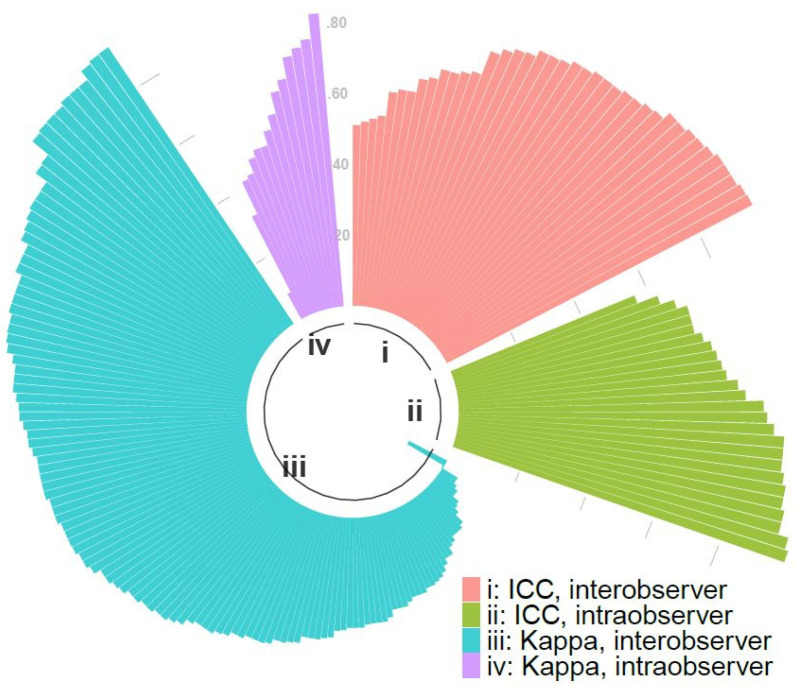
Circle bar chart visualizing the distribution of Kappa statistics and ICC values for every comparison.

**Figure 4 diagnostics-12-02400-f004:**
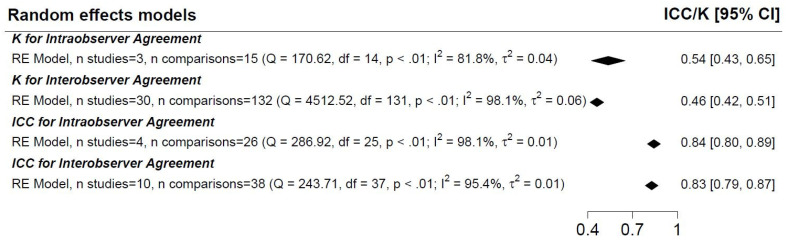
Pooled random effects model for inter/intra-observer agreement by studies reporting Kappa statistics or inter/intra-class correlation coefficient.

**Figure 5 diagnostics-12-02400-f005:**
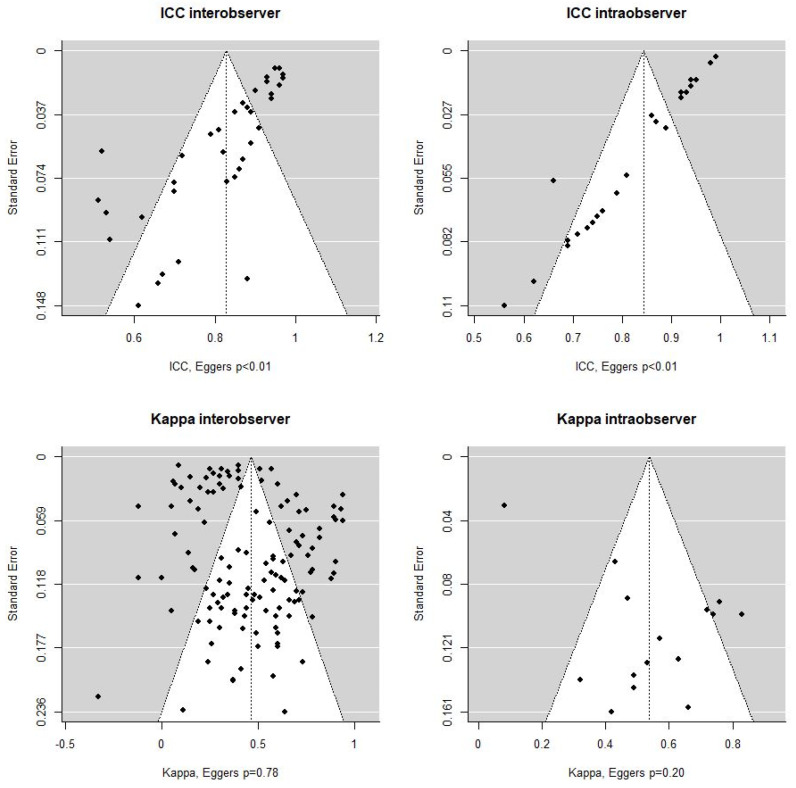
Eggers tests for inter/intra-observer agreements (ICC and Kappa models).

**Table 1 diagnostics-12-02400-t001:** Evaluation guideline.

Kappa		Intra-Class Correlation		Spearman Rank Correlation	
Value	Evaluation	Value	Evaluation	Value	Evaluation
>0.90	Almost perfect	>0.9	Excellent	±1	Perfect
0.80–0.90	Strong	0.75–0.9	Good	±0.8–0.9	Very strong
0.60–0.79	Moderate	0.5–<0.75	Moderate	±0.6–0.7	Moderate
0.40–0.59	Weak	<0.5	Poor	±0.3–0.5	Fair
0.21–0.39	Minimal			±0.1–0.2	Poor
<0.20	None			0	None

**Table 2 diagnostics-12-02400-t002:** Characteristics of included studies, including methodological quality assessment.

Reference (Year)	Single or Multi Center Study	n Included for Review (Total)	Indication	Finding Group(s)	MINORS Score (0–14)
Adler DG (2004) [18]	Single	20 (20)	GI bleeding	Blood; Erosions/Ulcerations	11
Alageeli M (2020) [19]	Multi	25 (25)	GI bleeding, CD, screening for HPS	Cleanliness	11
Albert J (2004) [20]	Single	36 (36)	OGIB, suspected CD, suspected SB tumor, refractory sprue, FAP	Cleanliness	12
Arieira C (2019) [21]	Single	22 (22)	Known CD	IBD	8
Biagi F (2006) [22]	Multi	21 (32)	CeD, IBS, known CD	Villous atrophy	10
Blanco-Velasco G (2021) [23]	Single	100 (100)	IDA, GI bleeding, known CD, SB tumors, diarrhea	Blood; IBD; Blended outcomes	11
Bossa F (2006) [24]	Single	39 (41)	OGIB, HPS, known CD, CeD, diarrhea	Blood; Blended outcomes; Other lesions; Polyps; Erosions/Ulcerations; Angiodysplasias	8
Bourreille A (2006) [25]	Multi	32 (32)	IIleocolonic resection	Blended outcomes; Other lesions; Villous atrophy; Erosions/Ulcerations	12
Brotz C (2009) [26]	Single	40 (541)	GI bleeding, abdominal pain, diarrhea, anemia, follow-up of prior findings	Cleanliness	10
Buijs MM (2018) [27]	Single	42 (136)	CRC screening	Blended outcomes; Polyps; Cleanliness	13
Chavalitdhamrong D (2012) [28]	Multi	65 (65)	Portal hypertension	Other lesions	12
Chetcuti Zammit S (2021) [29]	Multi	300 (300)	CeD, seronegative villous atrophy	IBD; Villous atrophy; Erosions/Ulcerations; Blended outcomes	13
Christodoulou D (2007) [30]	Single	20 (20)	GI bleeding	Other lesions; Angiodysplasias; Polyps; Blood	11
Cotter J (2015) [31]	Single	70 (70)	Known CD	IBD	12
De Leusse A (2005) [32]	Single	30 (64)	GI bleeding	Blood; Angiodysplasias; Other lesions; Erosions/Ulcerations; Blended outcomes;	12
de Sousa Magalhães R (2021) [33]	Single	58 (58)	Incomplete colonoscopy	Cleanliness	11
Delvaux M (2008) [34]	Multi	96 (98)	Known or suspected esophageal disease	Blended outcomes	13
D’Haens G (2015) [35]	Multi	20 (40)	Known CD	IBD	11
Dray X (2021) [36]	Multi	155 (637)	OGIB	Cleanliness	12
Duque G (2012) [37]	Single	20 (20)	GI bleeding	Blended outcomes	11
Eliakim R (2020) [38]	Single	54 (54)	Known CD	IBD	11
Esaki M (2009) [39]	Single	75 (102)	OGIB, FAP, GI lymphoma, PJS, GIST, carcinoid tumor	Cleanliness	12
Esaki M (2019) [40]	Multi	50 (108)	Suspected CD	Other lesions; Erosions/Ulcerations	10
Ewertsen C (2006) [41]	Single	33 (34)	OGIB, carcinoid tumors, angiodysplasias, diarrhea, immune deficiency, diverticular disease	Blended outcomes	8
Gal E (2008) [42]	Single	20 (20)	Known CD	IBD	7
Galmiche JP (2008) [43]	Multi	77 (89)	GERD symptoms	Other lesions	12
Garcia-Compean D (2021) [44]	Single	22 (22)	SB angiodysplasias	Angiodysplasias; Blended outcomes	12
Ge ZZ (2006) [45]	Single	56 (56)	OGIB, suspected CD, abdominal pain, suspected SB tumor, FAP, diarrhea, sprue	Cleanliness	12
Girelli CM (2011) [46]	Single	25 (35)	Suspected submucosal lesion	Other lesions	12
Goyal J (2014) [47]	Single	34 (34)	NA	Cleanliness	11
Gupta A (2010) [48]	Single	20 (20)	PJS	Polyps	11
Gupta T (2011) [49]	Single	60 (60)	OGIB	Other lesions	12
Hong-Bin C (2013) [50]	Single	63 (63)	GI bleeding, abdominal pain, chronic diarrhea	Cleanliness	11
Jang BI (2010) [51]	Multi	56 (56)	NA	Blended outcomes	10
Jensen MD (2010) [52]	Single	30 (30)	Known or suspected CD	Other lesions; IBD; Blended outcomes	11
Lai LH (2006) [53]	Single	58 (58)	OGIB, known CD, abdominal pain	Blended outcomes	10
Lapalus MG (2009) [54]	Multi	107 (120)	Portal hypertension	Other lesions	11
Laurain A (2014) [55]	Multi	77 (80)	Portal hypertension	Other lesions	12
Laursen EL (2009) [56]	Single	30 (30)	NA	Blended outcomes	12
Leighton JA (2011) [57]	Multi	40 (40)	Healthy volunteers	Cleanliness	13
Murray JA (2008) [58]	Single	37 (40)	CeD	IBD; Villous atrophy	12
Niv Y (2005) [59]	Single	50 (50)	IDA, abdominal pain, known CD, CeD, GI lymphoma, SB transplant	Blended outcomes	11
Niv Y (2012) [60]	Multi	50 (54)	Known CD	IBD	13
Oliva S (2014) [61]	Single	29 (29)	UC	IBD	14
Oliva S (2014) [62]	Single	198 (204)	Suspected IBD, OGIB, other symptoms	Cleanliness	12
Omori T (2020) [63]	Single	20 (196)	Known CD	IBD	8
Park SC (2010) [64]	Single	20 (20)	GI bleeding, IDA, abdominal pain, diarrhea	Cleanliness; Blended outcomes	8
Petroniene R (2005) [65]	Single	20 (20)	CeD, villous atrophy	Villous atrophy	12
Pezzoli A (2011) [66]	Multi	75 (75)	NA	Blood; Blended outcomes	12
Pons Beltrán V (2011) [67]	Multi	31 (273)	GI bleeding, suspected CD	Cleanliness	14
Qureshi WA (2008) [68]	Single	18 (20)	BE	Other lesions	11
Ravi S (2022) [69]	Single	10 (22)	GI bleeding	Other lesions	14
Rimbaş M (2016) [70]	Single	64 (64)	SB ulcerations	IBD	12
Rondonotti E (2014) [71]	Multi	32 (32)	NA	Other lesions	11
Sciberras M (2022) [72]	Multi	100 (182)	Suspected submucosal lesion	Other lesions	10
Shi HY (2017) [73]	Single	30 (150)	UC	IBD; Blood; Erosions/Ulcerations	14
Triantafyllou K (2007) [74]	Multi	87 (87)	Diabetes mellitus	Cleanliness; Blended outcomes	11
Usui S (2014) [75]	Single	20 (20)	UC	IBD	9
Wong RF (2006) [76]	Single	19 (32)	FAP	Polyps	13
Zakaria MS (2009) [77]	Single	57 (57)	OGIB	Blended outcomes	9

Abbreviations: BE, Barrett’s esophagus; CD, Crohn’s disease; CeD, celiac disease; CRC, colorectal cancer; FAP, familial adenomatous polyposis; GERD, gastroesophageal reflux disease; GI, gastrointestinal; GIST, gastrointestinal stromal tumor; HPS; hereditary polyposis syndrome; IBS, irritable bowel syndrome; IDA, iron-deficiency anemia; NA, not available; OGIB, obscure gastrointestinal bleeding; PJS, Peutz–Jeghers syndrome; SB, small bowel; UC, ulcerative colitis.

**Table 3 diagnostics-12-02400-t003:** Overall means combined inter/intra-observer statistics values.

Test Statistic	Mean	CI 95%	Range	Comparisons, n (Inter/Intra)	Studies, n	Evaluation
Kappa	0.53	0.51; 0.55	−0.33; 1.0	424 (383/41)	46	Weak
ICC	0.81	0.78; 0.84	0.51; 1.0	73 (41/32)	11	Good
Spearman Rank	0.73	0.68; 0.78	0.30; 1.0	60 (60/0)	5	Moderate
Kendall’s coefficient	0.89	0.86; 0.92	0.77; 1.0	20 (18/2)	2	n too small
Kolmogorov–Smirnov	0.99	-	0.98; 1.0	2 (2/0)	1	n too small

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
