# Peer review of "Inter/Intra-Observer Agreement in Video-Capsule Endoscopy: Are We Getting It All Wrong? A Systematic Review and Meta-Analysis"

_diagnostics, 2022, doi:10.3390/diagnostics12102400_

Round 1
Reviewer 1 Report
In the present systematic review with meta-analysis about reader’s agreement in videocapsule endoscopy (VCE), Cortegoso Valdivia et al showed that most of studies reported a low inter- and intra-observer agreement and high heterogeneity among studies.
Some types of small bowel lesions are more prone to misdiagnosis. In this meta-analysis, several types of indications have been pooled and, therefore, this may explain the very high heterogeneity. Therefore I suggest, if possible, to perform sub-analyses by aggregating data/studies according to the different indication/finding of VCE.
Please update literature search (December 2021 is quite outdated).
Reviewer 2 Report
Well written study and meta-analysis. I do not have major comments, just two points that need clarification by the authors
1) What does the I-CARE group stand for? I know that there is an i-care study organized by GETAID for IBD patients, but I do not think this is the same group. Please clarify
2) Most of the readers do not know what inter and intra-obsever agreemnt mean. I think that the authors need to clarify these terms better in the Introduction or Methods and also distinguish better the two terms in the Discussion
Round 2
Reviewer 1 Report
Answers were fine